# THE GAN LANDSCAPE: LOSSES, ARCHITECTURES, REGULARIZATION, AND NORMALIZATION

## ABSTRACT

Generative adversarial networks (GANs) are a class of deep generative models which aim to learn a target distribution in an unsupervised fashion. While they were successfully applied to many problems, training a GAN is a notoriously challenging task and requires a significant amount of hyperparameter tuning, neural architecture engineering, and a non-trivial amount of "tricks". The success in many practical applications coupled with the lack of a measure to quantify the failure modes of GANs resulted in a plethora of proposed losses, regularization and normalization schemes, and neural architectures. In this work we take a sober view of the current state of GANs from a practical perspective. We reproduce the current state of the art and go beyond fairly exploring the GAN landscape. We discuss common pitfalls and reproducibility issues, open-source our code on Github, and provide pre-trained models on TensorFlow Hub.

## 1 INTRODUCTION

Deep generative models are a powerful class of unsupervised machine learning models. The power of these models was recently harnessed in a variety of applications, including image generation, learned compression, and domain transfer (Isola et al., 2017; Radford et al., 2016; Agustsson et al., 2018; Tschannen et al., 2018). Generative adversarial networks (Goodfellow et al., 2014) are one of the main approaches to learning such models in a fully unsupervised fashion. The GAN framework can be viewed as a two-player game where the first player, the *generator*, is learning to transform some simple input distribution (usually a standard multivariate Normal or uniform) to a distribution on the space of images, such that the second player, the *discriminator*, cannot tell whether the samples belong to the true distribution or were synthesized. Both players aim to minimize their own loss and the solution to the game is the Nash equilibrium where neither player can improve their loss unilaterally. This powerful framework can also be derived by minimizing a divergence between the model distribution and the true distribution (Nowozin et al., 2016; Arjovsky et al., 2017).

Training GANs involves solving a minimax problem over the parameters of the generator and the discriminator which are usually parameterized as deep convolutional neural networks. Consequently, this minimax problem is notoriously hard to solve in practice. As a result, a plethora of loss functions, regularization and normalization schemes, coupled with neural architecture choices, have been proposed (Goodfellow et al., 2014; Salimans et al., 2016; Miyato et al., 2018; Gulrajani et al., 2017; Arjovsky et al., 2017; Mao et al., 2016).

**Our contributions.** In this work we provide a thorough empirical analysis of these competing approaches, and help the researchers and practitioners navigate this space. We first define the GAN landscape – the set of loss functions, normalization and regularization schemes, and the most commonly used architectures. We explore this search space on several modern large-scale data sets by means of hyperparameter optimization, considering both "good" sets of hyperparameters reported in the literature, as well as ones obtained by Gaussian Process regression. By analyzing the impact of the loss function, we conclude that the non-saturating loss is sufficiently stable across data sets, architectures and hyperparameters. We then proceed to decompose the effect of various normalization and regularization schemes, as well as varying architectures. We show that both gradient penalty (Gulrajani et al., 2017) as well as spectral normalization (Miyato et al., 2018) are

useful in the context of high-capacity architectures. Finally, we discuss some common pitfalls, reproducibility issues, and practical considerations. We provide reference implementations, including training and evaluation code on Github[1] and provide pre-trained models on TensorFlow Hub.[2]

## 2 THE GAN LANDSCAPE

### 2.1 LOSS FUNCTIONS

Let $P$ denote the target (true) distribution and $Q$ the model distribution. Goodfellow et al. (2014) suggest two loss functions: the minimax GAN and the non-saturating (NS) GAN. In the former the discriminator minimizes the negative log-likelihood for the binary classification task. In the latter the generator maximizes the probability of generated samples being real. In this work we consider the non-saturating loss as it is known to outperform the minimax variant. The corresponding loss functions are $\mathcal{L}_D = -\mathbb{E}_{x \sim P}[\log(D(x))] - \mathbb{E}_{\hat{x} \sim Q}[\log(1 - D(\hat{x}))]$ and $\mathcal{L}_G = -\mathbb{E}_{\hat{x} \sim Q}[\log(D(\hat{x}))]$.

In Wasserstein GAN (WGAN) (Arjovsky et al., 2017) the authors propose to consider the Wasserstein divergence instead of the original Jensen-Shannon (JS). In particular, under the optimal discriminator, minimizing the proposed value function with respect to the generator minimizes the Wasserstein distance between $P$ and $Q$. The drawback is that one has to ensure a 1-Lipschitz discriminator due to exploited Kantorovich-Rubenstein duality. The corresponding loss functions are $\mathcal{L}_D = -\mathbb{E}_{x \sim P}[D(x)] + \mathbb{E}_{\hat{x} \sim Q}[D(\hat{x})]$ and $\mathcal{L}_G = -\mathbb{E}_{\hat{x} \sim Q}[D(\hat{x})]$.

Finally, we consider the least-squares loss (LS) which corresponds to minimizing the Pearson $\chi^2$ divergence between $P$ and $Q$ (Mao et al., 2016). The intuition is that this loss function is smooth and saturates slower than the sigmoid cross-entropy loss of the JS formulation. The corresponding loss functions are $\mathcal{L}_D = -\mathbb{E}_{x \sim P}[(D(x) - 1)^2] + \mathbb{E}_{\hat{x} \sim Q}[D(\hat{x})^2]$ and $\mathcal{L}_G = -\mathbb{E}_{\hat{x} \sim Q}[(D(\hat{x}) - 1)^2]$.

### 2.2 REGULARIZATION AND NORMALIZATION OF THE DISCRIMINATOR

**Gradient norm penalty.** In the context of Wasserstein GANs this penalty can be interpreted as a soft penalty for the violation of 1-Lipschitzness (WGAN GP) (Gulrajani et al., 2017). Hereby, the gradient is evaluated on a linear interpolation between training points and generated samples as a proxy to the optimal coupling. The gradient penalty can also be evaluated around the data manifold which encourages the discriminator to be piece-wise linear in that region (Dragan) (Kodali et al., 2017). However, the gradient norm penalty can be considered purely as a regularizer for the discriminator and it was shown that it can improve the performance for other losses (Fedus et al., 2018). Furthermore, the penalty can be scaled by the "confidence" of the discriminator in the context of f-divergences (Roth et al., 2017). A drawback of gradient penalty (GP) regularization scheme is that it can depend on the model distribution $Q$ which changes during training. One drawback of Dragan is that it is unclear to which extent the Gaussian assumption for the manifold holds. Finally, computing the gradient norms implies a non-trivial running time penalty – essentially doubling the running time. We also investigate the impact of a regularizer ubiquitous in supervised learning – the $L_2$ penalty on all the weights of the network.

**Discriminator normalization.** Normalizing the discriminator can be useful from both the optimization perspective (more efficient gradient flow, a more stable optimization), as well as from the representation perspective – the representation richness of the layers in a neural network depends on the spectral structure of the corresponding weight matrices (Miyato et al., 2018).

From the optimization point of view, several techniques have found their way into the GAN literature, namely batch normalization (BN) (Ioffe and Szegedy, 2015) and layer normalization (LN) (Ba et al., 2016). Batch normalization in the context of GANs was suggested by Denton et al. (2015) and further popularized by Radford et al. (2016). It normalizes the pre-activations of nodes in a layer to mean $\beta$ and standard deviation $\gamma$, where both $\beta$ and $\gamma$ are parameters learned for each node in the layer. The normalization is done on the batch level and for each node separately. In contrast, with Layer normalization, all the hidden units in a layer share the same normalization terms $\beta$ and $\gamma$, but different

---

[1]Link removed to preserve anonymity.
[2]Link removed to preserve anonymity.

samples are normalized differently (Ba et al., 2016). Layer normalization was first applied in the context of GANs in Gulrajani et al. (2017).

From the representation point of view, one has to consider the neural network as a composition of (possibly non-linear) mappings and analyze their spectral properties. In particular, for the discriminator to be a bounded linear operator it suffices to control the maximum singular value. This approach is followed in Miyato et al. (2018) where the authors suggest dividing each weight matrix, including the matrices representing convolutional kernels, by their spectral norm. Furthermore, the authors argue that a key advantage of spectral normalization over competing approaches is that it results in discriminators of higher rank.

## 2.3 GENERATOR AND DISCRIMINATOR ARCHITECTURE

We explore two classes of architectures in this study: deep convolutional generative adversarial networks (DCGAN) (Radford et al., 2016) and residual networks (ResNet) (He et al., 2016), both of which are ubiquitous in GAN research. Recently, Miyato et al. (2018) defined a variation of DCGAN, so called *SNDCGAN*. Apart from minor updates (cf. Section 4) the main difference to DCGAN is the use of an eight-layer discriminator network. The details of both networks are summarized in Table 3. The other architecture, *ResNet19*, is an architecture with five ResNet blocks in the generator and six ResNet blocks in the discriminator, that can operate on $128 \times 128$ images. We follow the ResNet setup from Miyato et al. (2018), with the small difference that we simplified the design of the discriminator. The detailed parameters of discriminator and generator are summarized in Table 4a and Table 4b. With this setup we were able to reproduce the current state of the art results. An ablation study on various ResNet modifications is available in the Appendix.

## 2.4 EVALUATION METRICS

We focus on several recently proposed metrics well suited to the image domain. For an in-depth overview of quantitative metrics we refer the reader to (Borji, 2018).

**Inception Score (IS)**. Proposed by Salimans et al. (2016), IS offers a way to quantitatively evaluate the quality of generated samples. Intuitively, the conditional label distribution of samples containing meaningful objects should have low entropy, and the variability of the samples should be high. which can be expressed as $\text{IS} = \exp(\mathbb{E}_{x \sim Q}[d_{KL}(p(y \mid x), p(y))])$. The authors found that this score is well-correlated with scores from human annotators. Drawbacks include insensitivity to the prior distribution over labels and not being a proper *distance*.

As an alternative Heusel et al. (2017) proposed the **Frechet Inception Distance (FID)**. Samples from $P$ and $Q$ are first embedded into a feature space (a specific layer of InceptionNet). Then, assuming that the embedded data follows a multivariate Gaussian distribution, the mean and covariance are estimated. Finally, the Fréchet distance between these two Gaussians is computed, i.e.

$$\text{FID} = ||\mu_x - \mu_y||_2^2 + \text{Tr}(\Sigma_x + \Sigma_y - 2(\Sigma_x \Sigma_y)^{\frac{1}{2}}),$$

where $(\mu_x, \Sigma_x)$, and $(\mu_y, \Sigma_y)$ are the mean and covariance of the embedded samples from $P$ and $Q$, respectively. The authors argue that FID is consistent with human judgment and more robust to noise than IS. Furthermore, the score is sensitive to the visual quality of generated samples – introducing noise or artifacts in the generated samples will reduce the FID. In contrast to IS, FID can detect intra-class mode dropping, i.e. a model that generates only one image per class can score a perfect IS, but will suffer from have a high FID (Lucic et al., 2018). Bińkowski et al. (2018) argued that FID has no unbiased estimator and suggest **Kernel Inception distance (KID)** instead. In Appendix B we empirically compare KID to FID and observe that both metrics are very strongly correlated (Spearman rank-order correlation coefficient of $0.994$ for LSUN-BEDROOM and $0.995$ for CELEBA-HQ-$128$ datasets). As a result we focus on FID as it is likely to result in the same ranking.

**Multi-scale Structural Similarity for Image Quality (MS-SSIM) and Diversity.** A critical issue in GANs are *mode collapse* and *mode-dropping* – failing to capture a mode, or low-diversity of generated samples from a given mode. The MS-SSIM score (Wang et al., 2003) is used for measuring the similarity of two images where higher MS-SSIM score indicates more similar images. Several recent works suggest using the average pairwise MS-SSIM score within a given class as a proxy for the diversity of generated samples (Odena et al., 2017; Fedus et al., 2018). The drawback of this

Table 1: Hyperparameter ranges used in this study. The Cartesian product of the fixed values suffices to uncover the existing results. Gaussian Process optimization in the bandit setting (Srinivas et al., 2010) is used to select good hyperparameter settings from the specified ranges.

(a) Fixed values

| PARAMETER | DISCRETE VALUE |
|---|---|
| Learning rate $\alpha$ | $\{0.0002, 0.0001, 0.001\}$ |
| Reg. strength $\lambda$ | $\{1, 10\}$ |
| $(\beta_1, \beta_2, n_{dis})$ | $\{(0.5, 0.900, 5), (0.5, 0.999, 1),$ $(0.5, 0.999, 5), (0.9, 0.999, 5)\}$ |

(b) Gaussian Process regression ranges

| PARAMETER | RANGE | LOG |
|---|---|---|
| Learning rate $\alpha$ | $[10^{-5}, 10^{-2}]$ | Yes |
| $\lambda$ for $L_2$ | $[10^{-4}, 10^1]$ | Yes |
| $\lambda$ for non-$L_2$ | $[10^{-1}, 10^2]$ | Yes |
| $\beta_1 \times \beta_2$ | $[0, 1] \times [0, 1]$ | No |

approach is that we do not know the class corresponding to the generated sample, so it is usually applied on one-class data sets, such as CELEBA-HQ-128. In this work we use the same setup as in Fedus et al. (2018). In particular, given a batch size $b$, we compute the average pairwise MS-SSIM score on 5 batches, of $5 \times b \times (b - 1)/2$ image pairs in total. We stress that the diversity should only be taken into account *together with the FID and IS metrics*.

## 2.5 DATA SETS

We consider three data sets, namely CIFAR10, CELEBA-HQ-128, and LSUN-BEDROOM. The LSUN-BEDROOM data set (Yu et al., 2015) contains slightly more than 3 million images[3]. We randomly partition the images into a train and test set whereby we use 30588 images as the test set. Secondly, we use the CELEBA-HQ data set of 30k images (Karras et al., 2018). We use the $128 \times 128 \times 3$ version obtained by running the code provided by the authors.[4] We use 3000 examples as the test set and the remaining examples as the training set. Finally, we also include the CIFAR10 data set which contains 70K images (32x32x3), partitioned into 60000 training instances and 10000 testing instances. The baseline FID scores are 12.6 for CELEBA-HQ-128, 3.8 for LSUN-BEDROOM, and 5.19 for CIFAR10. Details on FID computation are presented in Section 4.

## 2.6 EXPLORING THE GAN LANDSCAPE

The search space for GANs is prohibitively expensive: exploring all combinations of all losses, normalization and regularization schemes, and architectures is outside of the practical realm. Instead, in this study we analyze several slices of this tensor for each data set. In particular, to ensure that we can reproduce existing results, we perform a study over the subset of this tensor on CIFAR10. We then proceed to analyze the performance of these models across CELEBA-HQ-128 and LSUN-BEDROOM. In Section 3.1 we fix everything but the loss. In Section 3.2 we fix everything but the regularization and normalization scheme. Finally, in Section 3.3 we fix everything but the architecture. This allows us to decouple some of these design choices and provide some insight on what matters most.

As noted in Lucic et al. (2018), one major issue preventing further progress is the hyperparameter tuning – currently, the community has converged to a small set of parameter values which work on some data sets, and may completely fail on others. In this study we combine the best hyperparameter settings found in the literature (Miyato et al., 2018), and perform Gaussian Process regression in the bandit setting (Srinivas et al., 2010) to possibly uncover better hyperparameter settings. We then consider the top performing models and discuss the impact of the computational budget.

We summarize the fixed hyperparameter settings in Table 1a which contains the "good" parameters reported in recent publications (Fedus et al., 2018; Miyato et al., 2018; Gulrajani et al., 2017). In particular, we consider the cross product of these parameters to obtain 24 hyperparameter settings to reduce the bias. Finally, to provide a fair comparison, we perform Gaussian Process optimization in the bandit setting (Srinivas et al., 2010) on the parameter ranges provided in Table 1b. We run 12 rounds (i.e. we communicate with the oracle 12 times) of the optimization, each with a batch of 10 hyperparameter sets selected based on the FID scores from the results of the previous iterations.

---

[3]The images are preprocessed to $128 \times 128 \times 3$ using TensorFlow resize_image_with_crop_or_pad.

[4]Available online at https://github.com/tkarras/progressive_growing_of_gans.

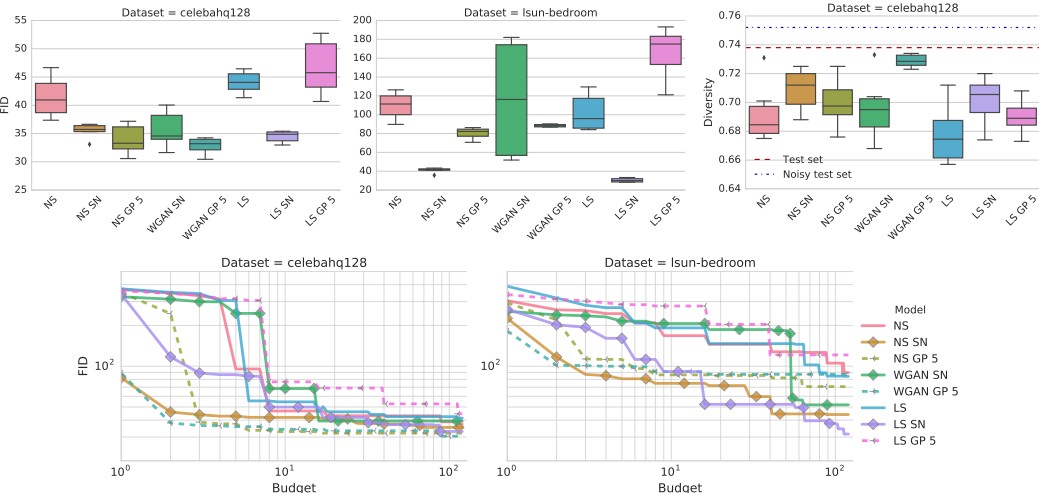

Figure 1: Impact of the loss function: FID distribution for top 5% models. The non-saturating (NS) loss is stable over both data sets. Gradient penalty and spectral normalization improve the sample quality. From the computational budget perspective (i.e. how many models one needs to train to reach a certain FID), both spectral normalization and gradient penalty perform better than the baseline, but the former is more efficient.

As we explore the number of discriminator updates per generator update (1 or 5), this leads to an additional 240 hyperparameter settings which in some cases outperform the previously known hyperparameter settings. Batch size is set to 64 for all the experiments. We use a fixed the number of discriminator update steps of 100K for LSUN-BEDROOM data set and CELEBA-HQ-128 data set, and 200K for CIFAR10 data set. We apply the Adam optimizer (Kingma and Ba, 2015).

## 3 RESULTS AND DISCUSSION

Given that there are 4 major components (loss, architecture, regularization, normalization) to analyze for each data set, it is infeasible to explore the whole landscape. Hence, we opt for a more pragmatic solution – we keep some dimensions fixed, and vary the others. For each experiment we highlight three aspects: (1) FID distribution of the top 5% of the trained models, (2) the corresponding sample diversity score, and (3) the tradeoff between the computational budget (i.e. number of models to train) and model quality in terms of FID. Each model was retrained 5 times with a different random seed and we report the median score. The variance for models obtained by Gaussian Process regression is handled implicitly so we train each model once.

### 3.1 IMPACT OF THE LOSS FUNCTION

Here the loss is either the non-saturating loss (NS) (Goodfellow et al., 2014), the least-squares loss (LS) (Mao et al., 2016), or the Wasserstein loss (WGAN) (Arjovsky et al., 2017). We use the ResNet19 with generator and discriminator architectures detailed in Table 4a. We consider the most prominent normalization and regularization approaches: gradient penalty (Gulrajani et al., 2017), and spectral normalization (Miyato et al., 2018). Both studies were performed on CELEBA-HQ-128 and LSUN-BEDROOM with hyperparameter settings shown in Table 1a.

The results are presented in Figure 1. We observe that the non-saturating loss is stable over both data sets. Spectral normalization improves the quality of the model on both data sets. Similarly, the gradient penalty can help improve the quality of the model, but finding a good regularization tradeoff is non-trivial and requires a high computational budget. Models using the GP penalty benefit from 5:1 ratio of discriminator to generator updates as suggested by (Gulrajani et al., 2017). We also performed a study on hinge loss (Miyato et al., 2018) and present it in the Appendix.

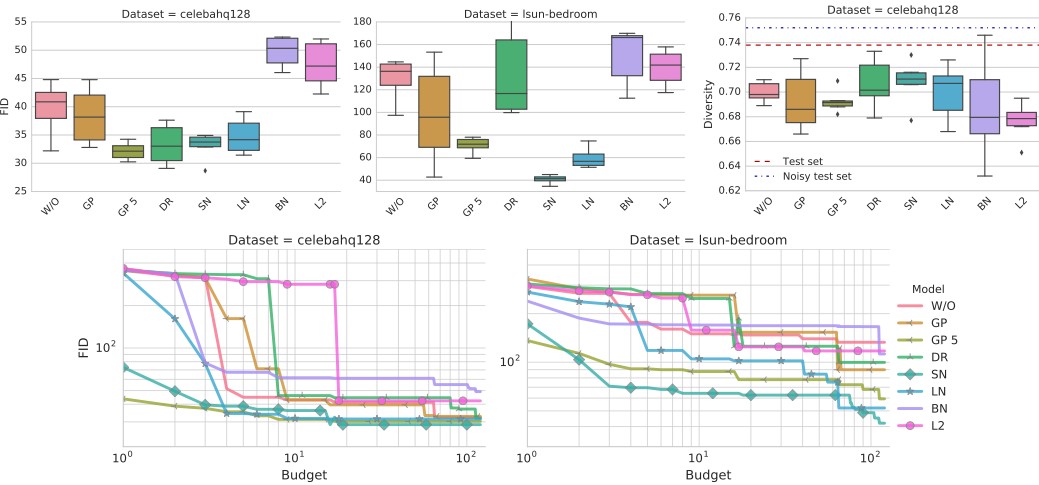

Figure 2: Impact of regularization and normalization: FID distribution for top 5% models. Both gradient penalty (GP) and spectral normalization (SN) outperform the baseline and should be considered, while former being more computationally expensive. Unfortunately none fully address the stability issues.

## 3.2 IMPACT OF REGULARIZATION AND NORMALIZATION

The goal of this study is to compare the relative performance of various regularization and normalization methods presented in the literature. To this end, and based on the loss study, we fix the loss to non-saturating loss (Goodfellow et al., 2014). We use the ResNet19 with generator and discriminator architectures described in Table 4a. Finally, we consider batch normalization (BN) (Ioffe and Szegedy, 2015), layer normalization (LN) (Ba et al., 2016), spectral normalization (SN), gradient penalty (GP) (Gulrajani et al., 2017), dragan penalty (DR) (Kodali et al., 2017), or $L_2$ regularization. We consider both CELEBA-HQ-128 and LSUN-BEDROOM with the hyperparameter settings shown in Table 1a and Table 1b.

The results are presented in Figure 2. We observe that adding batch norm to the discriminator hurts the performance. Secondly, gradient penalty can help, but it doesn't stabilize the training. In fact, it is non-trivial to strike a balance of the loss and regularization strength. Spectral normalization helps improve the model quality and is more computationally efficient than gradient penalty. This is consistent with recent results in Zhang et al. (2018). Similarly to the loss study, models using GP penalty benefit from 5:1 ratio of discriminator to generator updates. Furthermore, in a separate ablation study we observed that running the optimization procedure for an additional 100K steps is likely to increase the performance of the models with GP penalty.

**Impact of Simultaneous Regularization and Normalization.** Given the folklore that the Lipschitz constant of the discriminator is critical for the performance, one may expect simultaneous

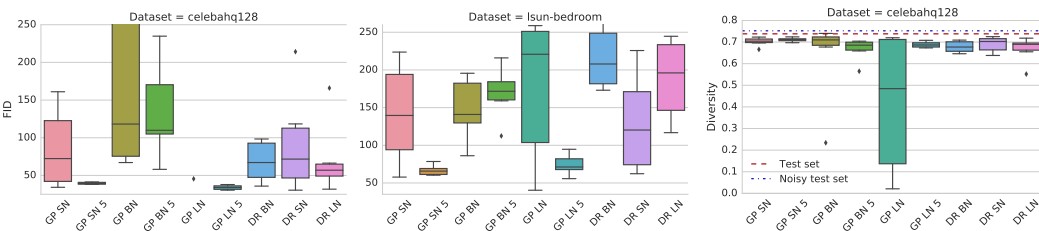

Figure 3: Impact of simultaneous normalization and regularization: FID distribution for top 5% models. Gradient penalty coupled with spectral normalization (SN) or layer normalization (LN) strongly improves the performance over the baseline.

regularization and normalization could improve model quality. To quantify this effect, we fix the loss to non-saturating loss (Goodfellow et al., 2014), use the Resnet19 architecture (as above), and combine several normalization and regularization schemes, with hyperparameter settings shown in Table 1a coupled with 24 randomly selected parameters. The results are presented in Figure 3. We observe that one may benefit from additional regularization and normalization. However, a lot of computational effort has to be invested for somewhat marginal gains in FID. Nevertheless, given enough computational budget we advocate simultaneous regularization and normalization – spectral normalization and layer normalization seem to perform well in practice.

### 3.3 IMPACT OF GENERATOR AND DISCRIMINATOR ARCHITECTURES

An interesting practical question is whether our findings also hold for a different model capacity. To this end, we also perform a study on SNDCGAN from Miyato et al. (2018). We consider the non-saturating GAN loss, gradient penalty and spectral normalization. While for smaller architectures regularization is not essential (Lucic et al., 2018), the regularization and normalization effects might become more relevant due to deeper architectures and optimization considerations.

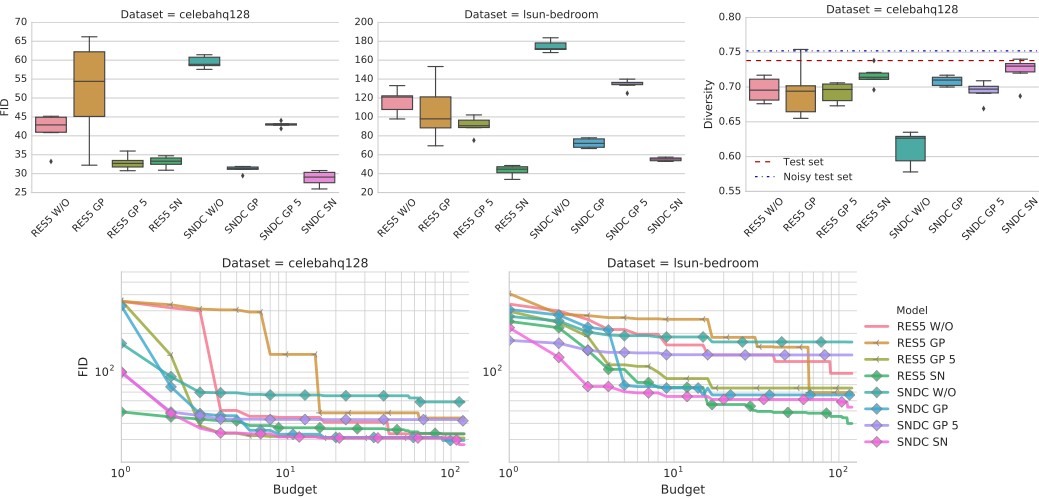

Figure 4: Impact of the neural architectures: FID distribution for top 5% models. Both spectral normalization and gradient penalty can help improve upon the non-regularized baseline.

The results are presented in Figure 4. We observe that both architectures achieve comparable results and benefit from regularization and normalization. Spectral normalization strongly outperforms the baseline for both architectures.

## 4 COMMON PITFALLS

In this section we focus on several pitfalls we encountered while trying to reproduce existing results and provide a fairly and accurate comparison.

**Metrics.** There already seems to be a divergence in how the FID score is computed: (1) Some authors report the score on training data, yielding a FID between 50k training and 50k generated samples (Unterthiner et al., 2018). Some opt to report the FID based on 10k test samples and 5k generated samples and use a custom implementation (Miyato et al., 2018). Finally, Lucic et al. (2018) report the score with respect to the test data, in particular FID between 10k test samples, and 10k generated samples. The subtle differences will result in a mismatch between the reported FIDs, in some cases of more than 10%. We argue that FID should be computed with respect to the test data set as and use 10k test samples and 10k generated samples on CIFAR10 and LSUN-BEDROOM, and 3k vs 3k on CELEBA-HQ-128 as in in Lucic et al. (2018). Similarly, there are several ways to compute a diversity score using MS-SSIM and we follow the approach from Fedus et al. (2018). We provide the implementation details in Section G of the Appendix.

**Details of neural architectures.** Even in popular architectures, like ResNet, there is still a number of design decision one needs to make, that are often omitted from the reported results. Those include the exact design of the ResNet cell (order of layers, when is ReLu applied, when to upsample and downsample, how many filters to use). Some of these differences might lead to potentially unfair comparison. As a result, we suggest to use the architectures presented within this work as a solid baseline. An ablation study on various ResNet modifications is available in the Appendix.

**Data sets.** A common issue is related to data set processing – does LSUN-BEDROOM always correspond to the same data set? In most cases the precise algorithm for upscaling or cropping is not clear which introduces inconsistencies between results on the "same" data set.

**Implementation details and non-determinism.** One major issue is the mismatch between the algorithm presented in a paper and the code provided online. We are aware that there is an embarrassingly large gap between a good implementation and a bad implementation of a given model. Hence, when no code is available, one is forced to guess which modifications were done. Another particularly tricky issue is removing randomness from the training process. After one fixes the data ordering and the initial weights, obtaining the same score by training the same model twice is non-trivial due to randomness present in certain GPU operations (Chetlur et al., 2014). Disabling the optimizations causing the non-determinism often results in an order of magnitude running time penalty.

While each of these issues taken in isolation seems minor, they compound to create a mist which introduces friction in practical applications and the research process (Sculley et al., 2018).

## 5 RELATED WORK

A recent large-scale study on GANs and Variational Autoencoders was presented in Lucic et al. (2018). The authors consider several loss functions and regularizers, and study the effect of the loss function on the FID score, with low-to-medium complexity data sets (MNIST, CIFAR10, CELEBA), and a single (InfoGAN style) architecture. In this limited setting, the authors found that there is no statistically significant difference between recently introduced models and the original non-saturating GAN. A study of the effects of gradient-norm regularization in GANs was recently presented in Fedus et al. (2018). The authors posit that the gradient penalty can also be applied to the non-saturating GAN, and that, to a limited extent, it reduces the sensitivity to hyperparameter selection. In a recent work on spectral normalization, the authors perform a small study of the competing regularization and normalization approaches (Miyato et al., 2018). We are happy to report that we could reproduce these results and we present them in the Appendix.

Inspired by these works and building on the available open-source code from Lucic et al. (2018), we take one additional step in all dimensions considered therein: more complex neural architectures, more complex data sets, and more involved regularization and normalization schemes.

## 6 CONCLUSION

In this work we study the GAN landscape: losses, regularization and normalization schemes, and neural architectures, and their impact on the on the quality of generated samples which we assess by recently introduced quantitative metrics. Our fair and thorough empirical evaluation suggests that one should consider non-saturating GAN loss and spectral normalization as default choices when applying GANs to a new data set. Given additional computational budget, we suggest adding the gradient penalty from Gulrajani et al. (2017) and train the model until convergence. Furthermore, additional marginal gains can be obtained by combining normalization and regularization empirically confirming the importance of the Lipschitz constant of the discriminator. Furthermore, both types of architectures proposed up-to this point perform reasonably well. A separate ablation study uncovered that most of the tricks applied in the ResNet style architectures lead to marginal changes in the quality and should be avoided due to the high computational cost. As a result of this large-scale study we identify the common pitfalls standing in the way of accurate and fair comparison and propose concrete actions to demystify the future results – issues with metrics, data set preprocessing, non-determinism, and missing implementation details are particularly striking. We hope that this work, together with the open-sourced reference implementations and trained models, will serve as a solid baseline for future GAN research.

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

## A  FID AND INCEPTION SCORES ON CIFAR10

We present an empirical study with SNDCGAN and ResNet CIFAR architectures on CIFAR10 in figure 5 and figure 6. In addition to Section 3.1, we evaluate one more kind of loss on CIFAR10. Here **HG**, NS and WGAN stand for **hinge loss**, non saturating loss and Wasserstein loss respectively. We observe that hinge loss performs very similar to non-saturating loss.

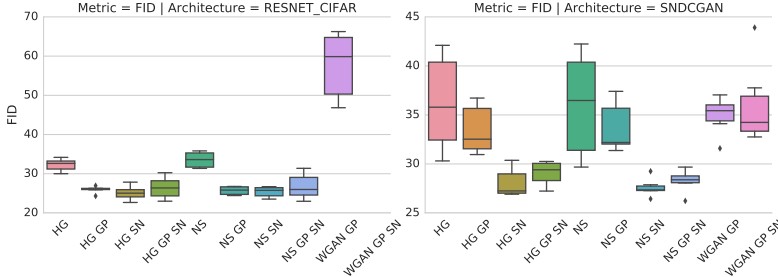

Figure 5: An empirical study with SNDCGAN and ResNet cifar architectures on CIFAR10. We recover the state of the art results recently reported in Miyato et al. (2018).

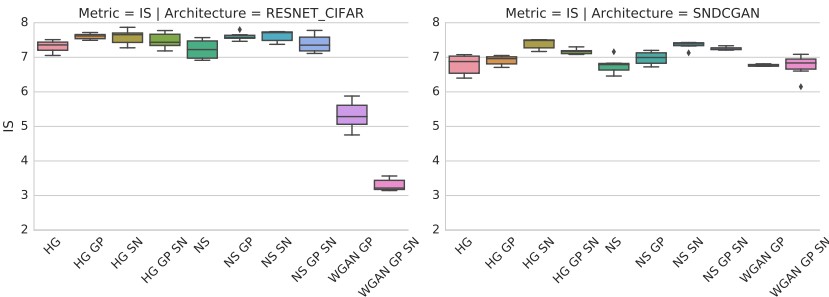

Figure 6: We show the Inception Score for each model within our study which corresponds to recently reported results (Miyato et al., 2018).

## B  COMPARISON OF FID AND KID METRICS

The KID metric introduced by Bińkowski et al. (2018) is an alternative to FID. We use models from our Regularization and Normalization study (see Section 3.2) to compare both metrics. Here, by model we denote everything that needs to be specified for the training – including all hyper-parameters, like learning rate, $\lambda$, Adam's $\beta$, etc. The Spearman rank-order correlation coefficient between KID and FID scores is approximately 0.994 for LSUN-BEDROOM and 0.995 for CELEBA-HQ-128 datasets.

To evaluate a practical setting of selecting several best models, we compare the intersection between the set of "best $K$ models by FID" and the set of "best $K$ models by KID" for $K \in 5, 10, 20, 50, 100$. The results are summarized in Table 2.

This experiment suggests that FID and KID metrics are very strongly correlated, and for the practical applications one can choose either of them. Also, the conclusions from our studies based on FID should transfer to studies based on KID.

Table 2: Intersection between set of top $K$ experiments selected by FID and KID metrics.

|  | LSUN-BEDROOM | CELEBA-HQ-128 |
| --- | --- | --- |
| K = 5 | 4/5 | 2/5 |
| K = 10 | 9/10 | 8/10 |
| K = 20 | 18/20 | 15/20 |
| K = 50 | 49/50 | 46/50 |
| K = 100 | 95/100 | 98/100 |

# C   ARCHITECTURES

## C.1   SNDCGAN

We used the same architecture as Miyato et al. (2018), with the parameters copied from the GitHub page[5]. In Table 3a and Table 3b, we describe the operations in layer column with order. Kernel size is described in format $[filter\_h, filter\_w, stride]$, input shape is $h \times w$ and output shape is $h \times w \times channels$. The slopes of all lReLU functions are set to 0.1. The input shape $h \times w$ is $128 \times 128$ for CELEBA-HQ-128 and LSUN-BEDROOM, $32 \times 32$ for CIFAR10.

Table 3: SNDCGAN architecture.

(a) SNDCGAN discriminator

| LAYER | KERNEL | OUTPUT |
|---|---|---|
| Conv, lReLU | $[3, 3, 1]$ | $h \times w \times 64$ |
| Conv, lReLU | $[4, 4, 2]$ | $h/2 \times w/2 \times 128$ |
| Conv, lReLU | $[3, 3, 1]$ | $h/2 \times w/2 \times 128$ |
| Conv, lReLU | $[4, 4, 2]$ | $h/4 \times w/4 \times 256$ |
| Conv, lReLU | $[3, 3, 1]$ | $h/4 \times w/4 \times 256$ |
| Conv, lReLU | $[4, 4, 2]$ | $h/8 \times w/8 \times 512$ |
| Conv, lReLU | $[3, 3, 1]$ | $h/8 \times w/8 \times 512$ |
| Linear | - | 1 |

(b) SNDCGAN generator

| LAYER | KERNEL | OUTPUT |
|---|---|---|
| $z$ | - | 128 |
| Linear, BN, ReLU | - | $h/8 \times w/8 \times 512$ |
| Deconv, BN, ReLU | $[4, 4, 2]$ | $h/4 \times w/4 \times 256$ |
| Deconv, BN, ReLU | $[4, 4, 2]$ | $h/2 \times w/2 \times 128$ |
| Deconv, BN, ReLU | $[4, 4, 2]$ | $h \times w \times 64$ |
| Deconv, Tanh | $[3, 3, 1]$ | $h \times w \times 3$ |

## C.2   RESNET ARCHITECTURE

The ResNet19 architecture is described in Table 4. RS column stands for the resample of the residual block, with downscale(D)/upscale(U)/none(-) setting. MP stands for mean pooling and BN for batch normalization. ResBlock is defined in Table 5. The addition layer merges two paths by adding them. The first path is a shortcut layer with exactly one convolution operation, while the second path consists of two convolution operations. The downscale layer and upscale layer are marked in Table 5. We used average pool with kernel $[2, 2, 2]$ for downscale, after the convolution operation. We used unpool from https://github.com/tensorflow/tensorflow/issues/2169 for upscale, before convolution operation. $h$ and $w$ are the input shape to the ResNet block, output shape depends on the RS parameter. $c_i$ and $c_o$ are the input channels and output channels for a ResNet block. Table 6 described the ResNet CIFAR architecture we used in Figure 5 for reproducing the existing results. Note that RS is set to none for third ResBlock and fourth ResBlock in discriminator. In this case, we used the same ResNet block defined in Table 5 without resampling.

---

[5]https://github.com/pfnet-research/chainer-gan-lib

Table 4: ResNet 19 architecture corresponding to "resnet_small" in https://github.com/pfnet-research/sngan_projection.

(a) ResNet19 discriminator

| LAYER | KERNEL | RS | OUTPUT |
|---|---|---|---|
| ResBlock | $[3, 3, 1]$ | D | $64 \times 64 \times 64$ |
| ResBlock | $[3, 3, 1]$ | D | $32 \times 32 \times 128$ |
| ResBlock | $[3, 3, 1]$ | D | $16 \times 16 \times 256$ |
| ResBlock | $[3, 3, 1]$ | D | $8 \times 8 \times 256$ |
| ResBlock | $[3, 3, 1]$ | D | $4 \times 4 \times 512$ |
| ResBlock | $[3, 3, 1]$ | D | $2 \times 2 \times 512$ |
| ReLU, MP | - | - | 512 |
| Linear | - | - | 1 |

(b) ResNet19 generator

| LAYER | KERNEL | RS | OUTPUT |
|---|---|---|---|
| $z$ | - | - | 128 |
| Linear | - | - | $4 \times 4 \times 512$ |
| ResBlock | $[3, 3, 1]$ | U | $8 \times 8 \times 512$ |
| ResBlock | $[3, 3, 1]$ | U | $16 \times 16 \times 256$ |
| ResBlock | $[3, 3, 1]$ | U | $32 \times 32 \times 256$ |
| ResBlock | $[3, 3, 1]$ | U | $64 \times 64 \times 128$ |
| ResBlock | $[3, 3, 1]$ | U | $128 \times 128 \times 64$ |
| BN, ReLU | - | - | $128 \times 128 \times 64$ |
| Conv | $[3, 3, 1]$ | - | $128 \times 128 \times 3$ |
| Sigmoid | - | - | $128 \times 128 \times 3$ |

Table 5: ResNet block definition.

(a) ResBlock discriminator

| LAYER | KERNEL | RS | OUTPUT |
|---|---|---|---|
| Shortcut | $[3, 3, 1]$ | D | $h/2 \times w/2 \times c_o$ |
| BN, ReLU | - | - | $h \times w \times c_i$ |
| Conv | $[3, 3, 1]$ | - | $h \times w \times c_o$ |
| BN, ReLU | - | - | $h \times w \times c_o$ |
| Conv | $[3, 3, 1]$ | D | $h/2 \times w/2 \times c_o$ |
| Addition | - | - | $h/2 \times w/2 \times c_o$ |

(b) ResBlock generator

| LAYER | KERNEL | RS | OUTPUT |
|---|---|---|---|
| Shortcut | $[3, 3, 1]$ | U | $2h \times 2w \times c_o$ |
| BN, ReLU | - | - | $h \times w \times c_i$ |
| Conv | $[3, 3, 1]$ | U | $2h \times 2w \times c_o$ |
| BN, ReLU | - | - | $2h \times 2w \times c_o$ |
| Conv | $[3, 3, 1]$ | - | $2h \times 2w \times c_o$ |
| Addition | - | - | $2h \times 2w \times c_o$ |

Table 6: ResNet CIFAR architecture.

(a) ResNet CIFAR discriminator

| LAYER | KERNEL | RS | OUTPUT |
|---|---|---|---|
| ResBlock | $[3, 3, 1]$ | D | $16 \times 16 \times 128$ |
| ResBlock | $[3, 3, 1]$ | D | $8 \times 8 \times 128$ |
| ResBlock | $[3, 3, 1]$ | - | $8 \times 8 \times 128$ |
| ResBlock | $[3, 3, 1]$ | - | $8 \times 8 \times 128$ |
| ReLU, MP | - | - | 128 |
| Linear | - | - | 1 |

(b) ResNet CIFAR generator

| LAYER | KERNEL | RS | OUTPUT |
|---|---|---|---|
| $z$ | - | - | 128 |
| Linear | - | - | $4 \times 4 \times 256$ |
| ResBlock | $[3, 3, 1]$ | U | $8 \times 8 \times 256$ |
| ResBlock | $[3, 3, 1]$ | U | $16 \times 16 \times 256$ |
| ResBlock | $[3, 3, 1]$ | U | $32 \times 32 \times 256$ |
| BN, ReLU | - | - | $32 \times 32 \times 256$ |
| Conv | $[3, 3, 1]$ | - | $32 \times 32 \times 3$ |
| Sigmoid | - | - | $32 \times 32 \times 3$ |

# D   RESNET ARCHITECTURE ABLATION STUDY

We have noticed six minor differences on Resnet architecture comparing to implementation from `https://github.com/pfnet-research/chainer-gan-lib/blob/master/common/net.py` (Miyato et al., 2018). We did ablation study to verify the impact of these differences. Figure 7 shows the impact of the ablation study, with details described as following.

- DEFAULT: ResNet CIFAR architecture with spectral normalization and non-saturating GAN loss.
- SKIP: Use input as output for the shortcut connection in the discriminator ResBlock. By default it was a conv layer with 3x3 kernel.
- CIN: Use $c_i$ for the discriminator ResBlock hidden layer output channels. By default it was $c_o$ in our setup, while Miyato et al. (2018) used $c_o$ for first ResBlock and $c_i$ for the rest.
- OPT: Use an optimized setup for the first discriminator ResBlock, which includes: (1) no ReLU, (2) a conv layer for the shortcut connections, (3) use $c_o$ instead of $c_i$ in ResBlock.
- CIN OPT: Use CIN and OPT together. It means the first ResBlock is optimized while the remaining ResBlocks use $c_i$ for the hidden output channels.
- SUM: Use reduce sum for the discriminator output. By default it was reduce mean.
- TAN: Use tanh for the generator output, as well as range [-1, 1] for discriminator input. By default it was sigmoid and discriminator input range $[0, 1]$.
- EPS: Use a bigger epsilon $2e-5$ for generator batch normalization. By default it was $1e-5$ in TensorFlow.
- ALL: Apply all the above differences together.

In the ablation study, the CIN experiment obtained the worst FID score. Combining with OPT, the CIN results were improved to the same level as the others which is reasonable because the first block has three input channels, which becomes a bottleneck for the optimization. Hence, using OPT and CIN together performs as well as the others. Overall, the impact of these differences are minor according to the study on CIFAR10.

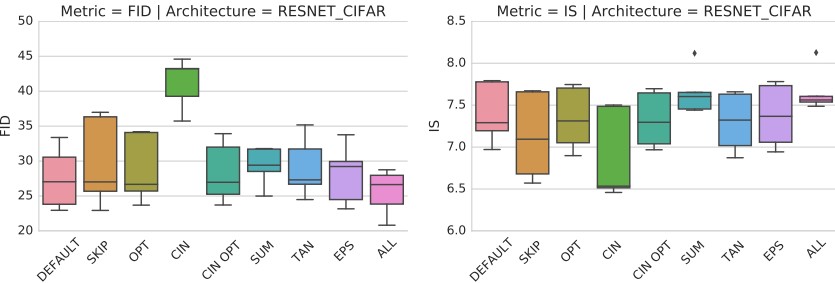

Figure 7: Ablation study of ResNet architecture differences. The experiment codes are described in Section D.

# E   RECOMMENDED HYPERPARAMETER SETTINGS

To make the future GAN training simpler, we propose a set of best parameters for three setups: (1) Best parameters without any regularizer. (2) Best parameters with only one regularizer. (3) Best parameters with at most two regularizers. Table 7, Table 8 and Table 9 summarize the top 2 parameters for SNDCGAN architecture, ResNet19 architecture and ResNet CIFAR architecture, respectively. Models are ranked according to the median FID score of five different random seeds with fixed hyper-parameters in Table 1a. Note that ranking models according to the best FID score of different seeds will achieve better but unstable result. Gaussian Process optimization hyper-parameters are not included in this table. For ResNet19 architecture with at most two regularizers, we have run it only once due to computational overhead. To show the model stability, we listed the best FID score out of five seeds from the same parameters in column best. Spectral normalization is clearly outperforms the other normalizers on SNDCGAN and ResNet CIFAR architectures, while on ResNet19 both layer normalization and spectral normalization work well.

To visualize the FID score on each data set, Figure 8, Figure 9 and Figure 10 show the generated examples by GANs. We select the examples from the best FID run, and then increase the FID score for two more plots.

Table 7: SNDCGAN parameters

| DATA SET | MEDIAN | BEST | LR($\times 10^{-3}$) | $\beta_1$ | $\beta_2$ | $n_{disc}$ | $\lambda$ | NORM |
|---|---|---|---|---|---|---|---|---|
| CIFAR10 | 29.75 | 28.66 | 0.100 | 0.500 | 0.999 | 1 | - | - |
| CIFAR10 | 36.12 | 33.23 | 0.200 | 0.500 | 0.999 | 1 | - | - |
| CELEBA-HQ-128 | 66.42 | 63.13 | 0.100 | 0.500 | 0.999 | 1 | - | - |
| CELEBA-HQ-128 | 67.39 | 64.59 | 0.200 | 0.500 | 0.999 | 1 | - | - |
| LSUN-BEDROOM | 180.36 | 160.12 | 0.200 | 0.500 | 0.999 | 1 | - | - |
| LSUN-BEDROOM | 188.99 | 162.00 | 0.100 | 0.500 | 0.999 | 1 | - | - |
| CIFAR10 | 26.66 | 25.27 | 0.200 | 0.500 | 0.999 | 1 | - | SN |
| CIFAR10 | 27.32 | 26.97 | 0.100 | 0.500 | 0.999 | 1 | - | SN |
| CELEBA-HQ-128 | 31.14 | 29.05 | 0.200 | 0.500 | 0.999 | 1 | - | SN |
| CELEBA-HQ-128 | 33.52 | 31.92 | 0.100 | 0.500 | 0.999 | 1 | - | SN |
| LSUN-BEDROOM | 63.46 | 58.13 | 0.200 | 0.500 | 0.999 | 1 | - | SN |
| LSUN-BEDROOM | 74.66 | 59.94 | 1.000 | 0.500 | 0.999 | 1 | - | SN |
| CIFAR10 | 26.23 | 26.01 | 0.200 | 0.500 | 0.999 | 1 | 1 | SN+GP |
| CIFAR10 | 26.66 | 25.27 | 0.200 | 0.500 | 0.999 | 1 | - | SN |
| CELEBA-HQ-128 | 31.13 | 30.80 | 0.100 | 0.500 | 0.999 | 1 | 10 | GP |
| CELEBA-HQ-128 | 31.14 | 29.05 | 0.200 | 0.500 | 0.999 | 1 | - | SN |
| LSUN-BEDROOM | 63.46 | 58.13 | 0.200 | 0.500 | 0.999 | 1 | - | SN |
| LSUN-BEDROOM | 66.58 | 65.75 | 0.200 | 0.500 | 0.999 | 1 | 10 | GP |

Table 8: ResNet19 parameters

| DATA SET | MEDIAN | BEST | LR($\times 10^{-3}$) | $\beta_1$ | $\beta_2$ | $n_{disc}$ | $\lambda$ | NORM |
|---|---|---|---|---|---|---|---|---|
| CELEBA-HQ-128 | 43.73 | 39.10 | 0.100 | 0.500 | 0.999 | 5 | - | - |
| CELEBA-HQ-128 | 43.77 | 39.60 | 0.100 | 0.500 | 0.999 | 1 | - | - |
| LSUN-BEDROOM | 160.97 | 119.58 | 0.100 | 0.500 | 0.900 | 5 | - | - |
| LSUN-BEDROOM | 161.70 | 125.55 | 0.100 | 0.500 | 0.900 | 5 | - | - |
| CELEBA-HQ-128 | 32.46 | 28.52 | 0.100 | 0.500 | 0.999 | 1 | - | LN |
| CELEBA-HQ-128 | 40.58 | 36.37 | 0.200 | 0.500 | 0.900 | 1 | - | LN |
| LSUN-BEDROOM | 70.30 | 48.88 | 1.000 | 0.500 | 0.999 | 1 | - | SN |
| LSUN-BEDROOM | 73.84 | 60.54 | 0.100 | 0.500 | 0.900 | 5 | - | SN |
| CELEBA-HQ-128 | 29.13 | - | 0.100 | 0.500 | 0.900 | 5 | 1 | LN+DR |
| CELEBA-HQ-128 | 29.65 | - | 0.200 | 0.500 | 0.900 | 5 | 1 | GP |
| LSUN-BEDROOM | 55.72 | - | 0.200 | 0.500 | 0.900 | 5 | 1 | LN+GP |
| LSUN-BEDROOM | 57.81 | - | 0.100 | 0.500 | 0.999 | 1 | 10 | SN+GP |

# F  WHICH PARAMETERS REALLY MATTER?

For each architecture and hyper-parameter we estimate its impact on the final FID. Figure 11 presents heatmaps for hyperparameters, namely the learning rate, $\beta_1$, $\beta_2$, $n_{disc}$, and $\lambda$ for each combination of neural architecture and data set.

# G  VARIATIONS OF MS-SSIM

We used the MS-SSIM scorer from TensorFlow with default power_factors (Wang et al., 2003). Note that the default filter size for each scale layer is 11, the minimum image edge is $11 \times 2^4 = 176$. To adapt it to CELEBA-HQ-128 data set with size $128 \times 128$, we used the minimum of filter size 11 and image size in last scale layer to allow the computation followed the previous work (Fedus et al., 2018).

Table 9: ResNet CIFAR parameters

| DATA SET | MEDIAN | BEST | LR($\times10^{-3}$) | $\beta_1$ | $\beta_2$ | $n_{disc}$ | $\lambda$ | NORM |
|---|---|---|---|---|---|---|---|---|
| CIFAR10 | 31.40 | 28.12 | 0.200 | 0.500 | 0.999 | 5 | - | - |
| CIFAR10 | 33.79 | 30.08 | 0.100 | 0.500 | 0.999 | 5 | - | - |
| CIFAR10 | 23.57 | 22.91 | 0.200 | 0.500 | 0.999 | 5 | - | SN |
| CIFAR10 | 25.50 | 24.21 | 0.100 | 0.500 | 0.999 | 5 | - | SN |
| CIFAR10 | 22.98 | 22.73 | 0.200 | 0.500 | 0.999 | 1 | 1 | SN+GP |
| CIFAR10 | 23.57 | 22.91 | 0.200 | 0.500 | 0.999 | 5 | - | SN |

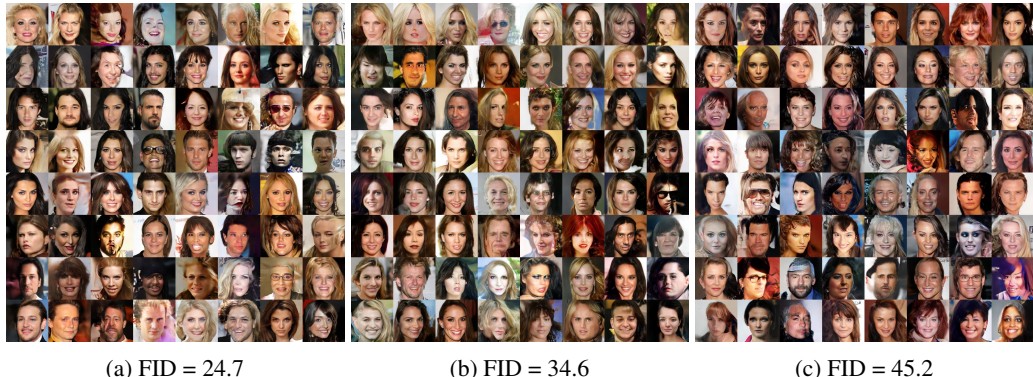

(a) FID = 24.7     (b) FID = 34.6     (c) FID = 45.2

Figure 8: Examples generated by GANs on CELEBA-HQ-128 data set.

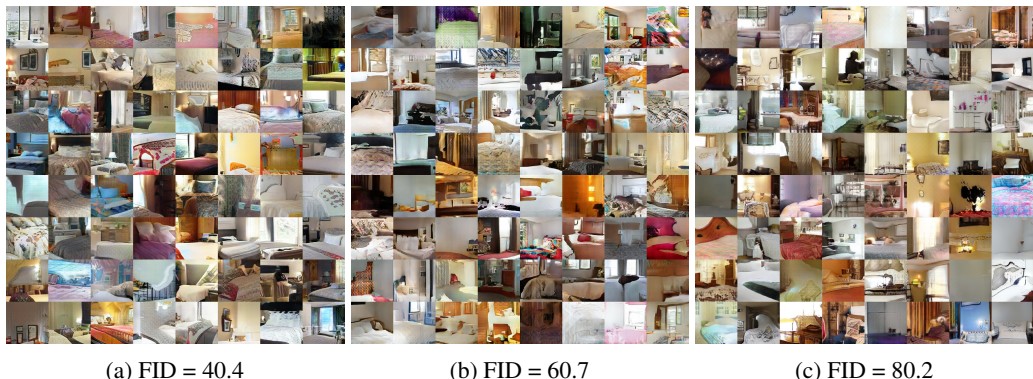

(a) FID = 40.4     (b) FID = 60.7     (c) FID = 80.2

Figure 9: Examples generated by GANs on LSUN-BEDROOM data set.

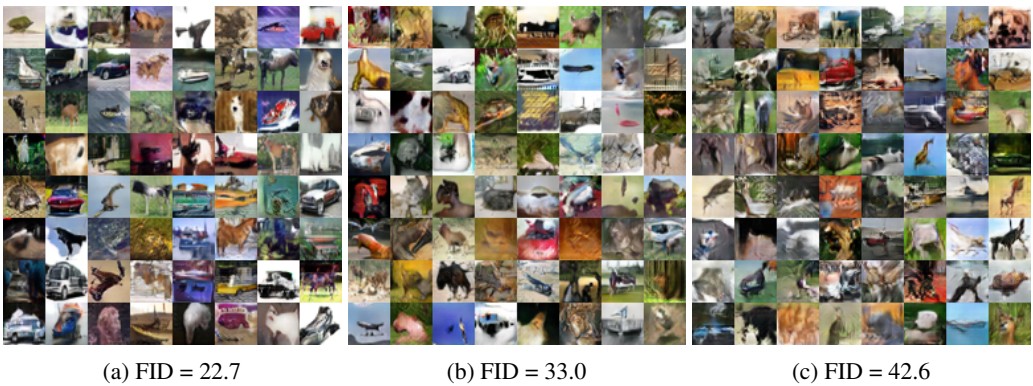

(a) FID = 22.7     (b) FID = 33.0     (c) FID = 42.6

Figure 10: Examples generated by GANs on CIFAR10 data set.

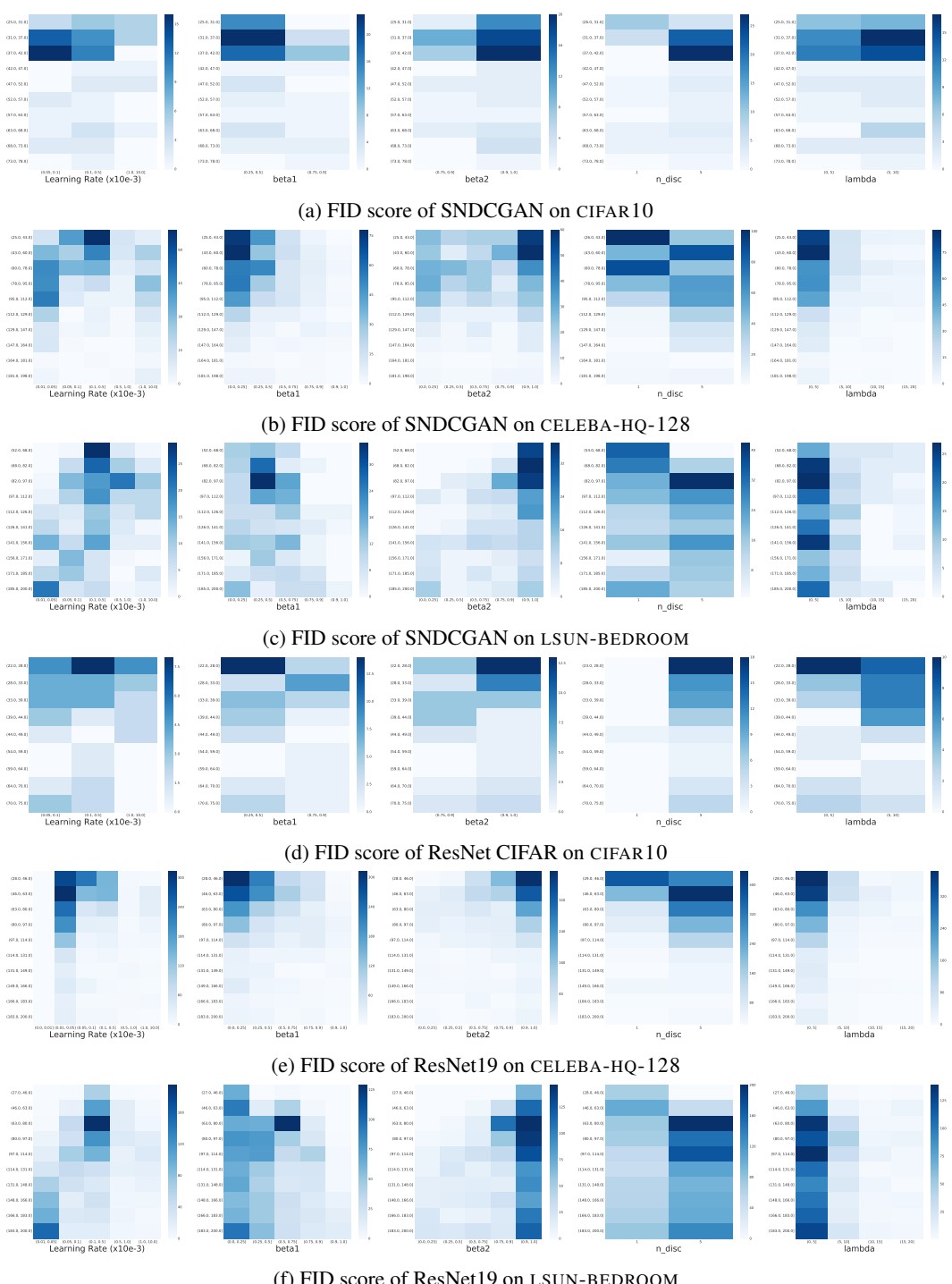

(a) FID score of SNDCGAN on CIFAR10

(b) FID score of SNDCGAN on CELEBA-HQ-128

(c) FID score of SNDCGAN on LSUN-BEDROOM

(d) FID score of ResNet CIFAR on CIFAR10

(e) FID score of ResNet19 on CELEBA-HQ-128

(f) FID score of ResNet19 on LSUN-BEDROOM

Figure 11: Heat plots for hyper-parameters on each architecture and dataset combination.

