# OpenReview forum: "The GAN Landscape: Losses, Architectures, Regularization, and Normalization"
_ICLR.cc/2019/Conference_

### Official Review · AnonReviewer3 · 2018-11-02
**A study of the effects of normalization and regularization in GANs**

**Rating:** 7
**Confidence:** 4

**Review:**

This paper seems to be an exposition on the primary performance affecting aspects of generative adversarial networks (GANs).  This can possibly affect our understanding of GANs, helping practitioners get the most in their applications, and perhaps leading to innovations that positively affect GAN performance.

Normally, expositions such as this I find difficult to recommend for publication. In these times, one can find "best practices" with a reasonable amount of rigor on data science blogs and such. An exposition that I would recommend for publication, would need to exhibit a high sense of depth and rigor for me to deem it publication worthy. This paper, for me, achieves this level of quality.

The authors start off by giving a precise, constrained list of hyperparameters and architectural components that they would explore. This is listed in the title and explained in detail in the beginning of the paper. The authors are right in explaining that they could not cover all hyperparameters and chose what I feel are quite salient ones. My one ask would have been a survey of how activations might affect performance. I sense that everyone has settled upon LeakyReLUs for internal layers, but a survey of that work and experimentation within the authors' framework would have been nice.

The authors then explain the metrics for evaluation and datasets. The datasets offered a healthy variety for typical image recognition tasks. It would be interesting to see what these metrics would reveal when applied to other types of data (e.g. scientific images).

The  authors explain, with graphs, the results of the loss, normalization, and architectures. I feel the discussion on loss was rushed, and I gained no insight on what the authors thought was a prominent difference between the three losses studied. Perhaps the authors had no salient observations for loss, but explicitly stating such would be useful to the reader. The only observation I gained as far as this is that non-saturating loss would possibly be stable across various datasets.

Regularization and normalization are discussed in much more detail, and I think the authors made helpful and interesting observations, such as the benefits of spectral normalization and the fact that batch normalization in the discriminator might be a harmful thing. These are good takeaways that could be useful to a vast number of GANs researchers.

For architectures to be a main pillar of the paper, I feel that this area could have been explored in greater detail. I feel that this discussion devolved into a discussion, again, about normalization rather than the architectural differences in performance. Unless I am misunderstanding something, it seems that the authors simply tested one more architecture, for the express purpose of testing whether their observations about normalization would hold.

As a bonus, the authors bring up some problems they had in making comparisons and reproducing results. I think this is an extremely important discussion to have, and I am glad that the authors detailed the obstacles in their journey. Hopefully this will inspire other researchers to avoid adding to the complications in this field.

The graphs were difficult to parse. I was able to make them out, but perhaps separating the top row (FID and diversity graphs) into separate figures, separate lines, or something would have reduced some confusion. In addition, different charts presenting only one loss function, with their spectral normalization and gradient penalty variants, would have made the effects of the normalization more obvious on the FID distribution graphs. If this can be changed before publication, I would strongly suggest it.

I appreciate that the authors provided source code via GitHub. However, in the future, the authors should be careful to provide an anonymous repository for review purposes. I had to be careful not to allow myself to focus on the author names which are prominent in the repository readme, and one of whom has his/her name in the GitHub URL itself. I didn't immediately recognize the names and thus it was easy for me not to retain them or focus on them. However, if it had been otherwise, it might have risked biasing the review.

In all, I think this is a good and useful paper from which I have learned and to which I will refer in the future as I continue my research into GANs and VAEs. I would suggest changing the title to be more appropriate and accurate (the researchers are primarily focused on showing the positive and negative effects of normalization across various loss functions and architectures). But altogether, I believe this is a paper worth publishing at ICLR.

---

> ### Author Response · Authors · 2018-11-13
> **Thank you for the actionable insights**
>
> [Q] My one ask would have been a survey of how activations might affect performance. I sense that everyone has settled upon LeakyReLUs for internal layers, but a survey of that work and experimentation within the authors' framework would have been nice.
> [A] We agree that this is an interesting question in it’s own right and should and will be explored more rigorously in future work. At this point, it seems like the number of parameters and whether skip-connections are used is much more impactful.
>
> [Q] It would be interesting to see what these metrics would reveal when applied to other types of data (e.g. scientific images).
> [A] We are aware of several works in the area of scientific images, such as [1] and [2], where GANs were successfully applied on 2D image snapshots from N-body simulations. The main issue for us at this point is having access to such data sets. Nevertheless, as these data sets become available for public, we will happily include them within our framework and investigate whether the conclusions extend to data sets beyond natural images.
> [1] https://arxiv.org/abs/1702.00403
> [2] https://arxiv.org/abs/1801.09070
>
> [Q] I feel the discussion on loss was rushed, and I gained no insight on what the authors thought was a prominent difference between the three losses studied.
> [A] The theoretical differences between these losses were studied in detail in the corresponding publications. From the practical side, it’s unclear which statistical divergence to optimize, in particular whether to pick (i) an f-divergence such as Chi-squared implemented by LS-GAN, or (ii) an integral probability metric such as Wasserstein distance, or (iii) a loss function which doesn’t correspond to any statistical divergence, such as NS-GAN. Hence, we wanted to provide some insight on how do these perform within different setups, not necessarily the ones used in the original publications. To this end we uncover that on the considered data sets it's hard to outperform the non-saturating loss combined with regularization and normalization. Apart from this, the empirical evidence doesn’t allow us to say more and we will clarify this in the manuscript.
>
> [Q] For architectures to be a main pillar of the paper, I feel that this area could have been explored in greater detail.
> [A] We agree with this assessment and we are indeed focusing on regularization and normalization. Our main question here was whether swapping Resnet with SNDCGAN leads to the same insights which is indeed the case. On the other hand, architectures are such a rich area enabling various design choices that they possibly merit a paper on their own. We will clarify the precise goal of the architecture exploration in this work. This being said, one major question we wanted to understand is which Resnet tricks from the literature (all 7 of them) are meaningful in practice and we present an ablation in the Section D of the appendix to conclude that the only relevant one is the number of channels which makes sense as it drastically changes the number of trainable parameters.
>
> [Q] The graphs were difficult to parse. I was able to make them out, but perhaps separating the top row (FID and diversity graphs) into separate figures, separate lines, or something would have reduced some confusion. In addition, different charts presenting only one loss function, with their spectral normalization and gradient penalty variants, would have made the effects of the normalization more obvious on the FID distribution graphs. If this can be changed before publication, I would strongly suggest it.
> [A] What we can do is to separate the top from the bottom figure into separate figures and provide more information in the captions. Furthermore, in Figure 1, for the FID distribution plots, we can group the methods visually (according to the loss function) by drawing a slightly shaded rectangle around results with the same loss (e.g. https://goo.gl/6YeUL1). If you have a specific proposal we would be happy to consider it and update the submission.
>
> [Q] In the future, the authors should be careful to provide an anonymous repository for review purposes.
> [A] This is a good point and we will address this issue in the future.
>
> [Q] I would suggest changing the title to be more appropriate and accurate (the researchers are primarily focused on showing the positive and negative effects of normalization across various loss functions and architectures).
> [A] Given the architecture discussion stated above, this is a valid point. Our current candidate is:
> “The GAN Landscape: The effect of Regularization and Normalization across various Losses and Neural Architectures”. However, if you have a specific proposal we would be happy to consider it.

---

### Official Review · AnonReviewer1 · 2018-11-02
**Okay contribution, but exposition could be better and lacks good take home messages**

**Rating:** 4
**Confidence:** 2

**Review:**

(As a disclamer I want to point out I'm not an expert in GANs and have only a basic understanding of the sub-field, but arguably this would make me target audience of this paper).

The authors presents a large scale study comparing a large number of GAN experiments, in this study they compare various choices of architechtures, losses and hyperparameters. The first part of the paper describes the various losses, architectures, regularization and normalization schemes; and the second part describes the results of the comparison experiments.

While I wish there were more such studies -- as I believe reproducing past results experimentally is important, and so is providing practical advice for practitioners -- this work in many parts hard to follow, and it is hard to get lot of new insight from the results, or a better understanding of GANs. As far I can see the most important take home message of the paper can be summarized in "one should consider non-saturating GAN loss and spectral normalization as default choices [...] Given additional computational budget, we suggest adding the
gradient penalty [...] and train the model until convergence".

Pros:
- available source code
- large number of experiments

Cons:
- the exposition could be improved, in particular the description of the plots is not very clear, I'm still not sure exactly what they show
- not clear what the target audience of the first part (section 2) is, it is too technical for a survey intended for outsiders, and discusses subtle points that are not easy to understand without more knowledge, but at the same time seems unlikely to give additional insight to an insider
- limited amount of new insight, which is limiting as new and better understanding of GANs and practical guidelines are arguably the main contribution of a work of this type


Some suggestions that I think could make the paper stronger

- I believe that in particular section 2 goes into too many mathematical details and subtleties that do not really add a lot. I think that either the reader already understand those concepts well (which I admit, I don't really, I'm merely curious about GANs and have been following the action from a distance, hence my low confidence rating to this review), or if they does not, it will be very hard to get much out of it. I would leave out some of the details, shortening the whole sections, and focus more on making a few of the concepts more understandable, and potentially leaving more space for a clearer description of the results
- it is not really clear to be what data the graphs show: the boxplots show 5% of what data? does it also include the models obtained by gaussian process regression? and what about the line plots, is it the best model so far as you train more and more models? if so, how are those models chosen and ordered? are they the results of single models or average of multiple ones?
- "the variance of models obtained by Guassian Process regression is handled implicitely so we tran each model once"? I do not understand what this means, and I work with hyper-parameter tuning using gaussian processes daily. It should probably be rephrased
- at the start of section 3: what is an "experiment"?
- in 3.1 towards the end of the first paragraph, what is a "study", is that the same as experiment or something different?
- (minor) stating that lower is better in the graphs might be useful
- (minor) typo in page 5 "We use a fixed the number"

---

> ### Author Response · Authors · 2018-11-13
> **Official response to AnonReviewer1**
>
> Thank you for the comments, please find our responses to specific points below.
>
> [Q] “As far as I can see the most important take home message of the paper can be summarized in "one should consider non-saturating GAN loss and spectral normalization as default choices [...] Given additional computational budget, we suggest adding the gradient penalty [...] and train the model until convergence."
> [A] While we want this study to be approachable by non-experts, some level of formalism is required as our main audience are researchers working on or interested in GANs. The summary you provided is indeed correct -- coupled with our open-sourced code, it allows a non-expert to train a GAN with state-of-the-art methods without needing to understand the details. On the other hand, for more experienced researchers, we provide more details on which design choices generalize to new settings and identify the biggest obstacles towards fair and unbiased quantitative evaluation of generative models.
>
> [Q] Limited amount of new insight.
> [A] Our paper presents many useful insights, namely: NS-GAN performs well, spectral norm is a good default normalization technique, gradient penalty should also be considered, even in combination with spectral norm but will cost substantially more in terms of computational resources, popular metrics such as KID and FID result in the same relative ordering of the models so there is no point in computing both, most Resnet tricks do not matter, etc. All of these insights are supported by a fair and unbiased rigorous experimental process. On top of that, our experiments are reproducible (as already reported by other works), we shared the resulting code and the pre-trained models.
>
> [Q] Clarification and exposition of plots.
> [A] Say that you had access to a GPU and had to train a model (loss+penalty+architecture). How many hyperparameter settings would you need to consider to achieve a certain quality? The FID from the plot is the estimate of the min FID computed by bootstrap estimation and the line-plots show this relationship. In other words, given a computing budget, which model should you pick? We will provide additional details in the caption of the plot.
>
> [Q] Bayesian optimization and variance.
> [A] We agree and will provide more details. When the sequential Bayesian optimization chooses the next set of hyperparameter combinations to test we run the model once (per hyperparameter combination) and report the scores to the optimizer. Then, the optimization algorithm takes these scores into account when selecting the next set of hyperparameters. The algorithm itself trades-off exploration and exploitation and it can explore hyperparameters "close" to the existing ones if they seem promising. Hence, the averaging happens implicitly during the search.
>
> [Q]: Studies and experiments. Stating that lower is better in the plots.
> [A]: Study is a set of experiments (say a study on the impact of the loss). Experiment is a concrete run with certain hyperparameters. Stating lower is better is a good idea, we will add this to the captions.

---

### Official Review · AnonReviewer2 · 2018-11-07
**An empirical study of GANs training techniques. Lacks significant novel insights**

**Rating:** 4
**Confidence:** 3

**Review:**


The paper studies several different techniques for training GANs: the architecture chosen, the loss function of the discriminator and generator,
and training techniques: normalization methods, ratio between updates of discriminator and generator, and regularization.
The method is performing an empirical training study on three image datasets, modifying the training procedure (e.g. changing one of the parameters) and using different metrics to evaluate the performance of the trained network.
Since the space of possible hyper-parameters , training algorithms, loss functions and network architecture is huge , the authors set a default training procedure, and in each numerical experiment freeze all techniques and parameters
except for one or two which they modify and evaluate.

The results of the paper do not give major insights into what are the preferred techniques for training GANs, and certainly not why and under what circumstances they'll work.
The authors recommend using non-saturated GANs loss and spectral normalization when training on new datasets, because these techniques achieved good performance metrics in most experiments.
But there is no attempt to generalize the findings (e.g. new datasets not from original study, changing other parameters and then evaluating again if these techniques help etc.), not clear if the
improvement in performance is statistically significant, how robust it is to changes in other parameters etc.
The authors also rely mostly on the FID metric, but do not show if and how there is improvement upon visual inspection of the generated images (i.e. is resolution improved, is fraction of images that look clearly 'unnatural' reduced etc.)

The writing is understandable for the most part, but the paper seems to lack focus - there is no clear take home message.
The authors use numerous jargon words to describe the techniques studied (e.g. dragon penalty, gradient penalty, spectral normalization, Gaussian process regression in the bandit setting) but they do not explain them,
give mathematical formulations, or insights into their advantages/disadvantages, making it hard to the non-expert reader to understand what are these techniques and why are they introduced.

With lack of clear novel insights, or at least more systematic study on additional datasets of the 'winning' techniques and a sensitivity analysis, the paper does not give a valuable enough contribution to the field to merit publication.

---

> ### Author Response · Authors · 2018-11-13
> **Official response to AnonReviewer2**
>
> Thank you for the time. We would like to take this opportunity to correct some factually incorrect statements below.
>
> [Q] The results of the paper do not give major insights into what are the preferred techniques for training GANs, and certainly not why and under what circumstances they'll work.
> [A]  We respectfully disagree. To our knowledge, this is only the second work which attempts to fairly and systematically compare GANs in a large-scale setting. The main conclusions of our work (about NS-GAN, spectral normalization, and gradient penalty) hold across several datasets and architectures.
>
> [Q] But there is no attempt to generalize the findings (e.g. new datasets not from original study, changing other parameters and then evaluating again if these techniques help etc.),
> [A] We again respectfully disagree -- both LSUN and CelebaHQ are used for the first time in such a large-scale evaluation. In fact, none of the techniques were previously evaluated on CelebaHQ. Furthermore, even if some data sets, such as LSUN, were used previously, the comparison to other works was always done by the authors of the new method usually with additional changes, such as architectural decisions and optimization tricks.
>
> [Q] Not clear if the improvement in performance is statistically significant, how robust it is to changes in other parameters etc.
> [A] In this we take care of systematically evaluating various design decisions. While the space of design decisions is too large to search over, we focus on the main design choices and provide some conclusions in this context. Performance improvements obtained by both spectral norm and gradient penalty are statistically significant as seen in the plots -- the performance with respect to the baseline is far outside of the two standard errors of the median in most settings.
>
> [Q] The authors also rely mostly on the FID metric, but do not show if and how there is improvement upon visual inspection of the generated images (i.e. is resolution improved, is fraction of images that look clearly 'unnatural' reduced etc.)
> [A] FID was shown to correlate well with perceived image quality (e.g. precision) and mode coverage (recall). The evidence can be found in [1], and [2]. As such, a reduction in FID corresponds both to improved image quality, as well as improved mode coverage. IN practice, a 10% drop in FID is visible to a human, and samples can be seen in the Appendix. While it is not a perfect metric, it is arguably useful for sample-based relative comparison of generative models.
> [1] https://arxiv.org/abs/1711.10337
> [2] https://arxiv.org/abs/1806.00035
>
> [Q] The authors use numerous jargon words to describe the techniques studied (e.g. dragon penalty, gradient penalty, spectral normalization, Gaussian process regression in the bandit setting) but they do not explain them, give mathematical formulations, making it hard to the non-expert reader to understand what are these techniques and why are they introduced.
> [A] Most of these are described in Section 2 (in particular, discussion on regularization and penalties is in Section 2.2). Describing all aspects of these techniques would require substantially more space and hence we refer to the original work for precise formulation.
>
> [Q] With lack of clear novel insights, or at least more systematic study on additional datasets of the 'winning' techniques and a sensitivity analysis, the paper does not give a valuable enough contribution to the field to merit publication.
> [A] We respectfully disagree: we believe that for GAN practitioners our paper presents many useful insights, namely: NS-GAN performs well, spectral norm is a good default normalization technique, gradient penalty should also be considered, even in combination with spectral norm but will cost substantially more in terms of computational resources, popular metrics such as KID and FID result in the same relative ordering of the models so there is no point in computing both, most resnet tricks do not matter, etc. All of these insights are supported by a fair and unbiased rigorous experimental process. On top of that, our experiments are reproducible (as already reported by other works), we shared the resulting code and the pre-trained models.

---

### Meta-Review · Area_Chair1 · 2018-12-08
**Intersting large scale empirical comparision that could profit from a clearer presentation.**

**Confidence:** 3
**Recommendation:** Reject

**Metareview:**

The paper presents a large scale empirical comparison between different prominent losses, regularization and normalization schemes, and neural architectures frequently used in GAN training. Large scale comparisons in this field are rare and important and the outcome of the experimental analysis is clearly of interest for practitioners. However, as two of the reviewers point out, the significance of the new insights is limited, and after rebutal all reviewers agree that the paper would profit from a clearer write-up and presentation of the main findings. I see the paper therefore, as lying slightly under the acceptance trashhold.